# Hindsight Experience Replay

**Marcin Andrychowicz**[*]**, Filip Wolski, Alex Ray, Jonas Schneider, Rachel Fong,
Peter Welinder, Bob McGrew, Josh Tobin, Pieter Abbeel**[†]**, Wojciech Zaremba**[†]
OpenAI

## Abstract

Dealing with sparse rewards is one of the biggest challenges in Reinforcement
Learning (RL). We present a novel technique called *Hindsight Experience Replay*
which allows sample-efficient learning from rewards which are sparse and binary
and therefore avoid the need for complicated reward engineering. It can be com-
bined with an arbitrary off-policy RL algorithm and may be seen as a form of
implicit curriculum.

We demonstrate our approach on the task of manipulating objects with a robotic
arm. In particular, we run experiments on three different tasks: pushing, sliding,
and pick-and-place, in each case using only binary rewards indicating whether or
not the task is completed. Our ablation studies show that Hindsight Experience
Replay is a crucial ingredient which makes training possible in these challenging
environments. We show that our policies trained on a physics simulation can
be deployed on a physical robot and successfully complete the task. The video
presenting our experiments is available at `https://goo.gl/SMrQnI`.

## 1  Introduction

Reinforcement learning (RL) combined with neural networks has recently led to a wide range of
successes in learning policies for sequential decision-making problems. This includes simulated
environments, such as playing Atari games (Mnih et al., 2015), and defeating the best human player
at the game of Go (Silver et al., 2016), as well as robotic tasks such as helicopter control (Ng et al.,
2006), hitting a baseball (Peters and Schaal, 2008), screwing a cap onto a bottle (Levine et al., 2015),
or door opening (Chebotar et al., 2016).

However, a common challenge, especially for robotics, is the need to engineer a reward function
that not only reflects the task at hand but is also carefully shaped (Ng et al., 1999) to guide the
policy optimization. For example, Popov et al. (2017) use a cost function consisting of five relatively
complicated terms which need to be carefully weighted in order to train a policy for stacking a
brick on top of another one. The necessity of cost engineering limits the applicability of RL in the
real world because it requires both RL expertise and domain-specific knowledge. Moreover, it is
not applicable in situations where we do not know what admissible behaviour may look like. It is
therefore of great practical relevance to develop algorithms which can learn from unshaped reward
signals, e.g. a binary signal indicating successful task completion.

One ability humans have, unlike the current generation of model-free RL algorithms, is to learn
almost as much from achieving an undesired outcome as from the desired one. Imagine that you are
learning how to play hockey and are trying to shoot a puck into a net. You hit the puck but it misses
the net on the right side. The conclusion drawn by a standard RL algorithm in such a situation would
be that the performed sequence of actions does not lead to a successful shot, and little (if anything)

---

[*] `marcin@openai.com`
[†] Equal advising.

would be learned. It is however possible to draw another conclusion, namely that this sequence of actions would be successful if the net had been placed further to the right.

In this paper we introduce a technique called *Hindsight Experience Replay (HER)* which allows the algorithm to perform exactly this kind of reasoning and can be combined with any off-policy RL algorithm. It is applicable whenever there are multiple *goals* which can be achieved, e.g. achieving each state of the system may be treated as a separate goal. Not only does HER improve the sample efficiency in this setting, but more importantly, it makes learning possible even if the reward signal is sparse and binary. Our approach is based on training universal policies (Schaul et al., 2015a) which take as input not only the current state, but also a goal state. The pivotal idea behind HER is to replay each episode with a different goal than the one the agent was trying to achieve, e.g. one of the goals which was achieved in the episode.

## 2 Background

### 2.1 Reinforcement Learning

We consider the standard reinforcement learning formalism consisting of an agent interacting with an environment. To simplify the exposition we assume that the environment is fully observable. An environment is described by a set of states $\mathcal{S}$, a set of actions $\mathcal{A}$, a distribution of initial states $p(s_0)$, a reward function $r : \mathcal{S} \times \mathcal{A} \to \mathbb{R}$, transition probabilities $p(s_{t+1}|s_t, a_t)$, and a discount factor $\gamma \in [0, 1]$.

A deterministic policy is a mapping from states to actions: $\pi : \mathcal{S} \to \mathcal{A}$. Every episode starts with sampling an initial state $s_0$. At every timestep $t$ the agent produces an action based on the current state: $a_t = \pi(s_t)$. Then it gets the reward $r_t = r(s_t, a_t)$ and the environment's new state is sampled from the distribution $p(\cdot|s_t, a_t)$. A discounted sum of future rewards is called a *return*: $R_t = \sum_{i=t}^{\infty} \gamma^{i-t} r_i$. The agent's goal is to maximize its expected return $\mathbb{E}_{s_0}[R_0|s_0]$. The Q-function or action-value function is defined as $Q^\pi(s_t, a_t) = \mathbb{E}[R_t|s_t, a_t]$.

Let $\pi^*$ denote an *optimal policy* i.e. any policy $\pi^*$ s.t. $Q^{\pi^*}(s, a) \geq Q^\pi(s, a)$ for every $s \in S, a \in A$ and any policy $\pi$. All optimal policies have the same Q-function which is called *optimal Q-function* and denoted $Q^*$. It is easy to show that it satisfies the following equation called the *Bellman* equation:

$$Q^*(s, a) = \mathbb{E}_{s' \sim p(\cdot|s,a)} \left[ r(s, a) + \gamma \max_{a' \in \mathcal{A}} Q^*(s', a') \right].$$

### 2.2 Deep Q-Networks (DQN)

*Deep Q-Networks (DQN)* (Mnih et al., 2015) is a model-free RL algorithm for discrete action spaces. Here we sketch it only informally, see Mnih et al. (2015) for more details. In DQN we maintain a neural network $Q$ which approximates $Q^*$. A *greedy* policy w.r.t. $Q$ is defined as $\pi_Q(s) = \text{argmax}_{a \in \mathcal{A}} Q(s, a)$. An $\epsilon$-greedy policy w.r.t. $Q$ is a policy which with probability $\epsilon$ takes a random action (sampled uniformly from $\mathcal{A}$) and takes the action $\pi_Q(s)$ with probability $1 - \epsilon$.

During training we generate episodes using $\epsilon$-greedy policy w.r.t. the current approximation of the action-value function $Q$. The transition tuples $(s_t, a_t, r_t, s_{t+1})$ encountered during training are stored in the so-called *replay buffer*. The generation of new episodes is interleaved with neural network training. The network is trained using mini-batch gradient descent on the loss $\mathcal{L}$ which encourages the approximated Q-function to satisfy the Bellman equation: $\mathcal{L} = \mathbb{E}(Q(s_t, a_t) - y_t)^2$, where $y_t = r_t + \gamma \max_{a' \in \mathcal{A}} Q(s_{t+1}, a')$ and the tuples $(s_t, a_t, r_t, s_{t+1})$ are sampled from the replay buffer[1].

### 2.3 Deep Deterministic Policy Gradients (DDPG)

*Deep Deterministic Policy Gradients (DDPG)* (Lillicrap et al., 2015) is a model-free RL algorithm for continuous action spaces. Here we sketch it only informally, see Lillicrap et al. (2015) for more details. In DDPG we maintain two neural networks: a *target policy* (also called an *actor*) $\pi : \mathcal{S} \to \mathcal{A}$ and an action-value function approximator (called the *critic*) $Q : \mathcal{S} \times \mathcal{A} \to \mathbb{R}$. The critic's job is to approximate the actor's action-value function $Q^\pi$.

Episodes are generated using a *behavioral policy* which is a noisy version of the target policy, e.g. $\pi_b(s) = \pi(s) + \mathcal{N}(0,1)$. The critic is trained in a similar way as the Q-function in DQN but the targets $y_t$ are computed using actions outputted by the actor, i.e. $y_t = r_t + \gamma Q(s_{t+1}, \pi(s_{t+1}))$. The actor is trained with mini-batch gradient descent on the loss $\mathcal{L}_a = -\mathbb{E}_s Q(s, \pi(s))$, where $s$ is sampled from the replay buffer. The gradient of $\mathcal{L}_a$ w.r.t. actor parameters can be computed by backpropagation through the combined critic and actor networks.

## 2.4 Universal Value Function Approximators (UVFA)

*Universal Value Function Approximators (UVFA)* (Schaul et al., 2015a) is an extension of DQN to the setup where there is more than one goal we may try to achieve. Let $\mathcal{G}$ be the space of possible goals. Every goal $g \in \mathcal{G}$ corresponds to some reward function $r_g : \mathcal{S} \times \mathcal{A} \to \mathbb{R}$. Every episode starts with sampling a state-goal pair from some distribution $p(s_0, g)$. The goal stays fixed for the whole episode. At every timestep the agent gets as input not only the current state but also the current goal $\pi : \mathcal{S} \times \mathcal{G} \to \mathcal{A}$ and gets the reward $r_t = r_g(s_t, a_t)$. The Q-function now depends not only on a state-action pair but also on a goal $Q^\pi(s_t, a_t, g) = \mathbb{E}[R_t | s_t, a_t, g]$. Schaul et al. (2015a) show that in this setup it is possible to train an approximator to the Q-function using direct bootstrapping from the Bellman equation (just like in case of DQN) and that a greedy policy derived from it can generalize to previously unseen state-action pairs. The extension of this approach to DDPG is straightforward.

# 3 Hindsight Experience Replay

## 3.1 A motivating example

Consider a bit-flipping environment with the state space $\mathcal{S} = \{0,1\}^n$ and the action space $\mathcal{A} = \{0,1,\ldots,n-1\}$ for some integer $n$ in which executing the $i$-th action flips the $i$-th bit of the state. For every episode we sample uniformly an initial state as well as a target state and the policy gets a reward of $-1$ as long as it is not in the target state, i.e. $r_g(s,a) = -[s \neq g]$.

Standard RL algorithms are bound to fail in this environment for $n > 40$ because they will never experience any reward other than $-1$. Notice that using techniques for improving exploration (e.g. VIME (Houthooft et al., 2016), count-based exploration (Ostrovski et al., 2017) or bootstrapped DQN (Osband et al., 2016)) does not help here because the real problem is *not* in lack of diversity of states being visited, rather it is simply impractical to explore such a large state space. The standard solution to this problem would be to use a shaped reward function which is more informative and guides the agent towards the goal, e.g. $r_g(s,a) = -||s - g||^2$. While using a shaped reward solves the problem in our toy environment, it may be difficult to apply to more complicated problems. We investigate the results of reward shaping experimentally in Sec. 4.4.

Figure 1: Bit-flipping experiment.

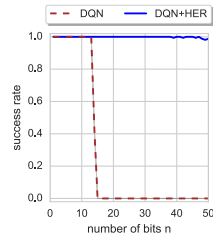

Instead of shaping the reward we propose a different solution which does not require any domain knowledge. Consider an episode with a state sequence $s_1, \ldots, s_T$ and a goal $g \neq s_1, \ldots, s_T$ which implies that the agent received a reward of $-1$ at every timestep. The pivotal idea behind our approach is to re-examine this trajectory with a different goal — while this trajectory may not help us learn how to achieve the state $g$, it definitely tells us something about how to achieve the state $s_T$. This information can be harvested by using an off-policy RL algorithm and experience replay where we replace $g$ in the replay buffer by $s_T$. In addition we can still replay with the original goal $g$ left intact in the replay buffer. With this modification at least half of the replayed trajectories contain rewards different from $-1$ and learning becomes much simpler. Fig. 1 compares the final performance of DQN with and without this additional replay technique which we call *Hindsight Experience Replay (HER)*. DQN without HER can only solve the task for $n \leq 13$ while DQN with HER easily solves the task for $n$ up to $50$. See Appendix A for the details of the experimental setup. Note that this approach combined with powerful function approximators (e.g., deep neural networks) allows the agent to learn how to achieve the goal $g$ even if it has never observed it during training.

We more formally describe our approach in the following sections.

## 3.2 Multi-goal RL

We are interested in training agents which learn to achieve multiple different goals. We follow the approach from *Universal Value Function Approximators* (Schaul et al., 2015a), i.e. we train policies

and value functions which take as input not only a state $s \in \mathcal{S}$ but also a goal $g \in \mathcal{G}$. Moreover, we show that training an agent to perform multiple tasks can be easier than training it to perform only one task (see Sec. 4.3 for details) and therefore our approach may be applicable even if there is only one task we would like the agent to perform (a similar situation was recently observed by Pinto and Gupta (2016)).

We assume that every goal $g \in \mathcal{G}$ corresponds to some predicate $f_g : \mathcal{S} \to \{0, 1\}$ and that the agent's goal is to achieve any state $s$ that satisfies $f_g(s) = 1$. In the case when we want to exactly specify the desired state of the system we may use $\mathcal{S} = \mathcal{G}$ and $f_g(s) = [s = g]$. The goals can also specify only some properties of the state, e.g. suppose that $\mathcal{S} = \mathbb{R}^2$ and we want to be able to achieve an arbitrary state with the given value of $x$ coordinate. In this case $\mathcal{G} = \mathbb{R}$ and $f_g((x, y)) = [x = g]$.

Moreover, we assume that given a state $s$ we can easily find a goal $g$ which is satisfied in this state. More formally, we assume that there is given a mapping $m : \mathcal{S} \to \mathcal{G}$ s.t. $\forall_{s \in \mathcal{S}} f_{m(s)}(s) = 1$. Notice that this assumption is not very restrictive and can usually be satisfied. In the case where each goal corresponds to a state we want to achieve, i.e. $\mathcal{G} = \mathcal{S}$ and $f_g(s) = [s = g]$, the mapping $m$ is just an identity. For the case of 2-dimensional state and 1-dimensional goals from the previous paragraph this mapping is also very simple $m((x, y)) = x$.

A universal policy can be trained using an arbitrary RL algorithm by sampling goals and initial states from some distributions, running the agent for some number of timesteps and giving it a negative reward at every timestep when the goal is not achieved, i.e. $r_g(s, a) = -[f_g(s) = 0]$. This does not however work very well in practice because this reward function is sparse and not very informative.

In order to solve this problem we introduce the technique of Hindsight Experience Replay which is the crux of our approach.

### 3.3 Algorithm

The idea behind Hindsight Experience Replay (HER) is very simple: after experiencing some episode $s_0, s_1, \ldots, s_T$ we store in the replay buffer every transition $s_t \to s_{t+1}$ not only with the original goal used for this episode but also with a subset of other goals. Notice that the goal being pursued influences the agent's actions but not the environment dynamics and therefore we can replay each trajectory with an arbitrary goal assuming that we use an off-policy RL algorithm like DQN (Mnih et al., 2015), DDPG (Lillicrap et al., 2015), NAF (Gu et al., 2016) or SDQN (Metz et al., 2017).

One choice which has to be made in order to use HER is the set of additional goals used for replay. In the simplest version of our algorithm we replay each trajectory with the goal $m(s_T)$, i.e. the goal which is achieved in the final state of the episode. We experimentally compare different types and quantities of additional goals for replay in Sec. 4.5. In all cases we also replay each trajectory with the original goal pursued in the episode. See Alg. 1 for a more formal description of the algorithm.

HER may be seen as a form of implicit curriculum as the goals used for replay naturally shift from ones which are simple to achieve even by a random agent to more difficult ones. However, in contrast to explicit curriculum, HER does not require having any control over the distribution of initial environment states. Not only does HER learn with extremely sparse rewards, in our experiments it also performs better with sparse rewards than with shaped ones (See Sec. 4.4). These results are indicative of the practical challenges with reward shaping, and that shaped rewards would often constitute a compromise on the metric we truly care about (such as binary success/failure).

## 4 Experiments

The video presenting our experiments is available at `https://goo.gl/SMrQnI`.

### 4.1 Environments

The are no standard environments for multi-goal RL and therefore we created our own environments. We decided to use manipulation environments based on an existing hardware robot to ensure that the challenges we face correspond as closely as possible to the real world. In all experiments we use a 7-DOF Fetch Robotics arm which has a two-fingered parallel gripper. The robot is simulated using the *MuJoCo* (Todorov et al., 2012) physics engine. The whole training procedure is performed in the simulation but we show in Sec. 4.6 that the trained policies perform well on the physical robot without any finetuning.

Policies are represented as Multi-Layer Perceptrons (MLPs) with Rectified Linear Unit (ReLU) activation functions. Training is performed using the *DDPG* algorithm (Lillicrap et al., 2015) with

---

**Algorithm 1** Hindsight Experience Replay (HER)

---
**Given:**
- an off-policy RL algorithm $\mathbb{A}$,                           ▷ e.g. DQN, DDPG, NAF, SDQN
- a strategy $\mathbb{S}$ for sampling goals for replay,             ▷ e.g. $\mathbb{S}(s_0, \ldots, s_T) = m(s_T)$
- a reward function $r : \mathcal{S} \times \mathcal{A} \times \mathcal{G} \to \mathbb{R}$.                           ▷ e.g. $r(s, a, g) = -[f_g(s) = 0]$

Initialize $\mathbb{A}$                           ▷ e.g. initialize neural networks
Initialize replay buffer $R$
**for** episode $= 1, M$ **do**
   Sample a goal $g$ and an initial state $s_0$.
   **for** $t = 0, T - 1$ **do**
      Sample an action $a_t$ using the behavioral policy from $\mathbb{A}$:
         $a_t \leftarrow \pi_b(s_t || g)$                           ▷ $||$ denotes concatenation
      Execute the action $a_t$ and observe a new state $s_{t+1}$
   **end for**
   **for** $t = 0, T - 1$ **do**
      $r_t := r(s_t, a_t, g)$
      Store the transition $(s_t || g, a_t, r_t, s_{t+1} || g)$ in $R$                           ▷ standard experience replay
      Sample a set of additional goals for replay $G := \mathbb{S}(\textbf{current episode})$
      **for** $g' \in G$ **do**
         $r' := r(s_t, a_t, g')$
         Store the transition $(s_t || g', a_t, r', s_{t+1} || g')$ in $R$                           ▷ HER
      **end for**
   **end for**
   **for** $t = 1, N$ **do**
      Sample a minibatch $B$ from the replay buffer $R$
      Perform one step of optimization using $\mathbb{A}$ and minibatch $B$
   **end for**
**end for**

---

*Adam* (Kingma and Ba, 2014) as the optimizer. See Appendix A for more details and the values of all hyperparameters.

We consider 3 different tasks:

1. *Pushing*. In this task a box is placed on a table in front of the robot and the task is to move it to the target location on the table. The robot fingers are locked to prevent grasping. The learned behaviour is a mixture of pushing and rolling.

2. *Sliding*. In this task a puck is placed on a long slippery table and the target position is outside of the robot's reach so that it has to hit the puck with such a force that it slides and then stops in the appropriate place due to friction.

3. *Pick-and-place*. This task is similar to pushing but the target position is in the air and the fingers are not locked. To make exploration in this task easier we recorded a *single* state in which the box is grasped and start half of the training episodes from this state[2].

The images showing the tasks being performed can be found in Appendix C.

**States:** The state of the system is represented in the MuJoCo physics engine.
**Goals:** Goals describe the desired position of the object (a box or a puck depending on the task) with some fixed tolerance of $\epsilon$ i.e. $\mathcal{G} = \mathbb{R}^3$ and $f_g(s) = [|g - s_{\textbf{object}}| \leq \epsilon]$, where $s_{\textbf{object}}$ is the position of the object in the state $s$. The mapping from states to goals used in HER is simply $m(s) = s_{\textbf{object}}$.
**Rewards:** Unless stated otherwise we use binary and sparse rewards $r(s, a, g) = -[f_g(s') = 0]$ where $s'$ if the state *after* the execution of the action $a$ in the state $s$. We compare sparse and shaped reward functions in Sec. 4.4.
**State-goal distributions:** For all tasks the initial position of the gripper is fixed, while the initial position of the object and the target are randomized. See Appendix A for details.
**Observations:** In this paragraph *relative* means relative to the *current* gripper position. The policy is

given as input the absolute position of the gripper, the relative position of the object and the target[3], as well as the distance between the fingers. The Q-function is additionally given the linear velocity of the gripper and fingers as well as relative linear and angular velocity of the object. We decided to restrict the input to the policy in order to make deployment on the physical robot easier.

**Actions:** None of the problems we consider require gripper rotation and therefore we keep it fixed. Action space is 4-dimensional. Three dimensions specify the desired relative gripper position at the next timestep. We use MuJoCo constraints to move the gripper towards the desired position but Jacobian-based control could be used instead[4]. The last dimension specifies the desired distance between the 2 fingers which are position controlled.

**Strategy $\mathbb{S}$ for sampling goals for replay:** Unless stated otherwise HER uses replay with the goal corresponding to the final state in each episode, i.e. $\mathbb{S}(s_0, \ldots, s_T) = m(s_T)$. We compare different strategies for choosing which goals to replay with in Sec. 4.5.

## 4.2 Does HER improve performance?

In order to verify if HER improves performance we evaluate DDPG with and without HER on all 3 tasks. Moreover, we compare against DDPG with count-based exploration[5] (Strehl and Littman, 2005; Kolter and Ng, 2009; Tang et al., 2016; Bellemare et al., 2016; Ostrovski et al., 2017). For HER we store each transition in the replay buffer twice: once with the goal used for the generation of the episode and once with the goal corresponding to the final state from the episode (we call this strategy `final`). In Sec. 4.5 we perform ablation studies of different strategies $\mathbb{S}$ for choosing goals for replay, here we include the best version from Sec. 4.5 in the plot for comparison.

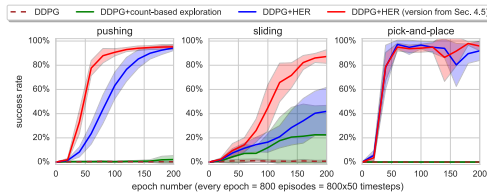
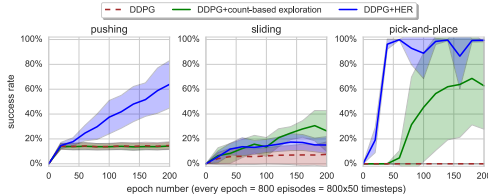

Figure 2: Multiple goals.

Figure 3: Single goal.

Fig. 2 shows the learning curves for all 3 tasks[6]. DDPG without HER is unable to solve any of the tasks[7] and DDPG with count-based exploration is only able to make some progress on the sliding task. On the other hand, DDPG with HER solves all tasks almost perfectly. It confirms that HER is a crucial element which makes learning from sparse, binary rewards possible.

## 4.3 Does HER improve performance even if there is only one goal we care about?

In this section we evaluate whether HER improves performance in the case where there is only one goal we care about. To this end, we repeat the experiments from the previous section but the goal state is identical in all episodes.

From Fig. 3 it is clear that DDPG+HER performs much better than pure DDPG even if the goal state is identical in all episodes. More importantly, comparing Fig. 2 and Fig. 3 we can also notice that HER learns faster if training episodes contain multiple goals, so in practice it is advisable to train on multiple goals even if we care only about one of them.

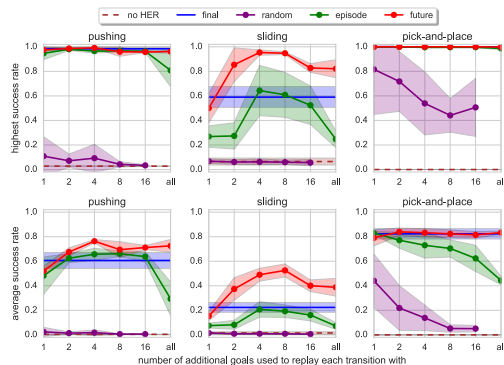

Figure 4: Ablation study of different strategies for choosing additional goals for replay. The top row shows the highest (across the training epochs) test performance and the bottom row shows the average test performance across all training epochs. On the right top plot the curves for `final`, `episode` and `future` coincide as all these strategies achieve perfect performance on this task.

## 4.4   How does HER interact with reward shaping?

So far we only considered binary rewards of the form $r(s, a, g) = -[|g - s_{\mathbf{object}}| > \epsilon]$. In this section we check how the performance of DDPG with and without HER changes if we replace this reward with one which is shaped. We considered reward functions of the form $r(s, a, g) = \lambda|g - s_{\mathbf{object}}|^p - |g - s'_{\mathbf{object}}|^p$, where $s'$ is the state of the environment after the execution of the action $a$ in the state $s$ and $\lambda \in \{0, 1\}$, $p \in \{1, 2\}$ are hyperparameters.

Surprisingly neither DDPG, nor DDPG+HER was able to successfully solve any of the tasks with any of these reward functions[8](learning curves can be found in Appendix D). Our results are consistent with the fact that successful applications of RL to difficult manipulation tasks which does not use demonstrations usually have more complicated reward functions than the ones we tried (e.g. Popov et al. (2017)).

The following two reasons can cause shaped rewards to perform so poorly: (1) There is a huge discrepancy between what we optimize (i.e. a shaped reward function) and the success condition (i.e.: is the object within some radius from the goal at the end of the episode); (2) Shaped rewards penalize for inappropriate behaviour (e.g. moving the box in a wrong direction) which may hinder exploration. It can cause the agent to learn not to touch the box at all if it can not manipulate it precisely and we noticed such behaviour in some of our experiments.

Our results suggest that domain-agnostic reward shaping does not work well (at least in the simple forms we have tried). Of course for every problem there exists a reward which makes it easy (Ng et al., 1999) but designing such shaped rewards requires a lot of domain knowledge and may in some cases not be much easier than directly scripting the policy. This strengthens our belief that learning from sparse, binary rewards is an important problem.

## 4.5   How many goals should we replay each trajectory with and how to choose them?

In this section we experimentally evaluate different strategies (i.e. $\mathbb{S}$ in Alg. 1) for choosing goals to use with HER. So far the only additional goals we used for replay were the ones corresponding to the final state of the environment and we will call this strategy `final`. Apart from it we consider the following strategies: `future` — replay with $k$ random states which come from the same episode as the transition being replayed and were observed *after* it, `episode` — replay with $k$ random states coming from the same episode as the transition being replayed, `random` — replay with $k$ random states encountered so far in the whole training procedure. All of these strategies have a hyperparameter $k$ which controls the ratio of HER data to data coming from normal experience replay in the replay buffer.

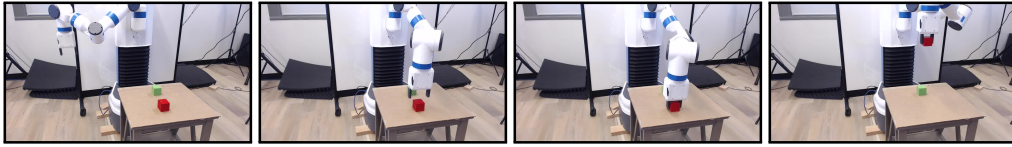

Figure 5: The pick-and-place policy deployed on the physical robot.

The plots comparing different strategies and different values of $k$ can be found in Fig. 4. We can see from the plots that all strategies apart from `random` solve pushing and pick-and-place almost perfectly regardless of the values of $k$. In all cases `future` with $k$ equal 4 or 8 performs best and it is the only strategy which is able to solve the sliding task almost perfectly. The learning curves for `future` with $k = 4$ can be found in Fig. 2. It confirms that the most valuable goals for replay are the ones which are going to be achieved in the near future[9]. Notice that increasing the values of $k$ above 8 degrades performance because the fraction of normal replay data in the buffer becomes very low.

### 4.6 Deployment on a physical robot

We took a policy for the pick-and-place task trained in the simulator (version with the `future` strategy and $k = 4$ from Sec. 4.5) and deployed it on a physical fetch robot without any finetuning. The box position was predicted using a separately trained CNN using raw fetch head camera images. See Appendix B for details.

Initially the policy succeeded in 2 out of 5 trials. It was not robust to small errors in the box position estimation because it was trained on perfect state coming from the simulation. After retraining the policy with gaussian noise (std=1cm) added to observations[10] the success rate increased to $5/5$. The video showing some of the trials is available at `https://goo.gl/SMrQnI`.

## 5 Related work

The technique of experience replay has been introduced in Lin (1992) and became very popular after it was used in the DQN agent playing Atari (Mnih et al., 2015). *Prioritized* experience replay (Schaul et al., 2015b) is an improvement to experience replay which prioritizes transitions in the replay buffer in order to speed up training. It it orthogonal to our work and both approaches can be easily combined.

Learning simultaneously policies for multiple tasks have been heavily explored in the context of policy search, e.g. Schmidhuber and Huber (1990); Caruana (1998); Da Silva et al. (2012); Kober et al. (2012); Devin et al. (2016); Pinto and Gupta (2016). Learning off-policy value functions for multiple tasks was investigated by Foster and Dayan (2002) and Sutton et al. (2011). Our work is most heavily based on Schaul et al. (2015a) who considers training a *single* neural network approximating multiple value functions. Learning simultaneously to perform multiple tasks has been also investigated for a long time in the context of Hierarchical Reinforcement Learning, e.g. Bakker and Schmidhuber (2004); Vezhnevets et al. (2017).

Our approach may be seen as a form of implicit curriculum learning (Elman, 1993; Bengio et al., 2009). While curriculum is now often used for training neural networks (e.g. Zaremba and Sutskever (2014); Graves et al. (2016)), the curriculum is almost always hand-crafted. The problem of automatic curriculum generation was approached by Schmidhuber (2004) who constructed an asymptotically optimal algorithm for this problem using program search. Another interesting approach is PowerPlay (Schmidhuber, 2013; Srivastava et al., 2013) which is a general framework for automatic task selection. Graves et al. (2017) consider a setup where there is a fixed discrete set of tasks and empirically evaluate different strategies for automatic curriculum generation in this settings. Another approach investigated by Sukhbaatar et al. (2017) and Held et al. (2017) uses self-play between the policy and a task-setter in order to automatically generate goal states which are on the border of what the current policy can achieve. Our approach is orthogonal to these techniques and can be combined with them.

# 6 Conclusions

We introduced a novel technique called Hindsight Experience Replay which makes possible applying RL algorithms to problems with sparse and binary rewards. Our technique can be combined with an arbitrary off-policy RL algorithm and we experimentally demonstrated that with DQN and DDPG.

We showed that HER allows training policies which push, slide and pick-and-place objects with a robotic arm to the specified positions while the vanilla RL algorithm fails to solve these tasks. We also showed that the policy for the pick-and-place task performs well on the physical robot without any finetuning. As far as we know, it is the first time so complicated behaviours were learned using only sparse, binary rewards.

**Acknowledgments**

We would like to thank Ankur Handa, Jonathan Ho, John Schulman, Matthias Plappert, Tim Salimans, and Vikash Kumar for providing feedback on the previous versions of this manuscript. We would also like to thank Rein Houthooft and the whole OpenAI team for fruitful discussions as well as Bowen Baker for performing some additional experiments.

## Footnotes

[1]The targets $y_t$ depend on the network parameters but this dependency is ignored during backpropagation. Moreover, DQN uses the so-called *target network* to make the optimization procedure more stable but we omit it here as it is not relevant to our results.

[2]This was necessary because we could not successfully train any policies for this task without using the demonstration state. We have later discovered that training is possible without this trick if only the goal position is sometimes on the table and sometimes in the air.

[3]The target position is relative to the current *object* position.

[4]The successful deployment on a physical robot (Sec. 4.6) confirms that our control model produces movements which are reproducible on the physical robot despite not being fully physically plausible.

[5] We discretize the state space and use an intrinsic reward of the form $\alpha/\sqrt{N}$, where $\alpha$ is a hyperparameter and $N$ is the number of times the given state was visited. The discretization works as follows. We take the relative position of the box and the target and then discretize every coordinate using a grid with a stepsize $\beta$ which is a hyperparameter. We have performed a hyperparameter search over $\alpha \in \{0.032, 0.064, 0.125, 0.25, 0.5, 1, 2, 4, 8, 16, 32\}$, $\beta \in \{1\text{cm}, 2\text{cm}, 4\text{cm}, 8\text{cm}\}$. The best results were obtained using $\alpha = 1$ and $\beta = 1\text{cm}$ and these are the results we report.

[6]An episode is considered successful if the distance between the object and the goal at the end of the episode is less than 7cm for pushing and pick-and-place and less than 20cm for sliding. The results are averaged across 5 random seeds and shaded areas represent one standard deviation.

[7]We also evaluated DQN (without HER) on our tasks and it was not able to solve any of them.

[8]We also tried to rescale the distances, so that the range of rewards is similar as in the case of binary rewards, clipping big distances and adding a simple (linear or quadratic) term encouraging the gripper to move towards the object but none of these techniques have led to successful training.

[9]We have also tried replaying the goals which are close to the ones achieved in the near future but it has not performed better than the `future` strategy

[10]The Q-function approximator was trained using exact observations. It does not have to be robust to noisy observations because it is not used during the deployment on the physical robot.

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
