[Reviews · NeurIPS 2017]

Reviewer 1



The main idea of the work is that it can be possible to replay an unsuccessful trajectory with a modification of the goal that it actually achieves. Overall, I'd say that it's not a huge/deep idea, but a very nice addition to the learning toolbox. When it is applicable, it seems like an idea that should be used. I worry that it is rarely applicable, though, without a tremendous amount of task insight. Consider the "kick the ball into the goal" example. If we want to speed up the learning of the task, we would likely need to use the idea to help guide the robot to the ball, to guide the robot's foot to make contact with the ball, and to guide the robot's aim to send the ball into the goal. At this level, it doesn't feel particularly different from (potential-based) reward shaping, which has been used in precisely this way. One claim could be that HER is a version of reward shaping that is more effective with current RL implementations (deep learning). I think that would be a much more useful comparison than the ones given in the paper. It puts the technique into context more completely. Note that there are small errors in the text should be fixed, for example: "The last dimension specify", "have lead", "can not", "the the most". "less than 20cm for sliding": Are the units right, there? I watched the video and it looked like it was placings things much much more closely than that. How big is 20cm relative to the size of the robot's workspace? The paper compares HER to an implicit curriculum. As I said above, I think it is more akin to reward shaping. But, in the curriculum space, I see it more as a method for generating "easy missions" from the existing ones. Again, overall, I like the idea and I can see some cases where it could be used. But, I think the paper doesn't put the technique into its proper context and that weakens the overall impact of the work. NOTE: The rebuttal says, "We agree that there is a connection between HER and reward shaping. Please note that experimental comparison of HER and reward shaping is included in Sec. 4.4 and shows that HER performs much better than reward shaping in our environments.". I do not think that this comment directly addresses my question, which was "At this level, it doesn't feel particularly different from (potential-based) reward shaping, which has been used in precisely this way.". That is, the shaping in 4.4 is not potential-based shaping. My sense (from trying to generalize your idea to more complex problems) is that you need to identify notions of subgoals to use your technique and you need to identify notions of subgoals to use potential based shaping. With those same subgoals identified, which technique is more successful? My concern is that the power of the algorithm in the demonstrations you showed is more about identifying subgoals than it is about how you are using them.

Reviewer 2



Summary: This paper introduces a method called hindsight experience replay (HER), which is designed to improve performance in sparse reward, RL tasks. The basic idea is to recognize that although a trajectory through the state-space might fail to find a particular goal, we can imagine that the trajectory ended at some other goal state-state. This simple idea when realized via the experience replay buffer of a deep q learning agent improves performance in several interesting simulated arm manipulation domains. Decision: I think this paper is an accept. The main idea is conceptually simple (in a good way). The paper is clear, and includes results on a toy domain to explain the idea, and then progresses to show good performance on simulated arm tasks. I have several questions that I invite the authors to respond to, in order to refine my final judgement of the paper. I will start with high-level questions. The auxiliary task work [1] (and hinted at with the Horde paper), shows that learning to predict and control many reward functions in parallel, results in improvement performance on the main task (for example the score in an atari game). One can view the auxiliary tasks as providing dense reward signals for training a deep network and thus regularizing the agent’s representation of the world to be useful for many task. In the auxiliary task work, pixel control tasks seems to improve performance the most, but the more general idea of feature control was also presented. I think the paper under review should discuss the connections with this work, and investigate what it might mean to use the auxiliary task approach in the simulated arm domain. Are both methods fundamentally getting at the same idea? What are the key differences? The work under-review is based on episodic tasks with goal states. How should we think about continuing tasks common in continuous control domains? Is the generalization straightforward under the average reward setting? The related work section mentions prioritized DQN is orthogonal. I don’t agree. Consider the possibility that prioritized DQN performs so well in all the domains you tested, that there remains no room for improvement when adding HER. I think this should be validated empirically. One could consider options as another approach to dealing with sparse reward domains: options allow the agent to jump through the state space, perhaps making it easier to find the goal state. Clearly there are challenges with discovering options, but the connection is worth mention. The result on the real robot, is too quickly discussed and doesn’t seem to well support the main ideas of the paper. It feels like: “lets add a robot result to make the paper appear stronger”. I suggest presenting a clearer result or dropping it and pulling in something else from the appendix. Footnote 3 says count-based exploration method was implemented by first discretizing the state. My understanding of Bellemare’s work was that it was introduced as a generalization of tabular exploration bonuses to the case of general function approximation. Was discretization required here, and if not what effect does it have on performance? The paper investigates the combination of HER with shaped rewards, showing the shaping actually hurts performance. This is indeed interesting, but a nice companion result would be investigating the combination of HER and shaping in a domain/task where we know shaping alone works well. [1] Jaderberg, M., Mnih, V., Czarnecki, W. M., Schaul, T., Leibo, J. Z., Silver, D., & Kavukcuoglu, K. (2016). Reinforcement learning with unsupervised auxiliary tasks. arXiv preprint arXiv:1611.05397.

Reviewer 3



In this paper, a new algorithm for RL is proposed, called "Hindsight Experience Replay" (HER). It is based on adding the description of the goal to be attained to the input of an off-policy algorithm like DQN or DDPG. It is then possible to augment the experience replay buffer with "fake" goals that were actually attained in the episode, in addition to the "true" goal we were trying to achieve. This helps converge when rewards are sparse, as it implicitly defines a form of curriculum where the agent first learns to reach goals that are easily attainable with random exploration, allowing it to progressively move towards more difficult goals. Experiments on robot manipulation tasks, where the goal is described by the target position of some object being manipulated (ex: pushing or grasping), show improvement over DDPG, count-based exploration and reward shaping, along with the ability to transfer a policy from a simulation environment to the real world. This is a great paper, extremely clear, with extensive experiments demonstrating the benefits of the method which, although very simple to understand and implement, is novel as far as I'm aware of, and may be an inspiration for further meaningful advances in RL. It also has the advantage of being "compatible" with many existing algorithms and their variants. I don't have much to say as I find the current submission to be very convincing already. My only high-level suggestion would be to make it clearer that: (1) HER does not technically *require* "sparse and binary" rewards, even if it may work better in some cases in such settings (2) (this is my opinion based on my understanding, if I am wrong it would be interesting to discuss it in more depth) HER is mostly meant to be used with a continuous "goal space", obtained from a mapping of the state space, such that knowledge of how to solve a goal can be re-used to solve a nearby goal (in this goal space). If that's not the case (ex: discrete goals represented by a one-one vector), there might still be some benefits to HER as extra goals may help find meaningful features, but it seems to me it won't transfer knowledge between goals as effectively, and in particular will still suffer if rewards for the goal of interest are very sparse. Note also that with discrete goals it gets pretty close to the algorithm from "Reinforcement Learning wih Unsupervised Auxiliary Tasks" (which could be cited here). Minor: - l.216: "specify" => specifies - "in practice it is advisable to train on multiple goals even if we care only about one of them": it does not seem so obvious to me, because the success rates in Fig. 2 and Fig. 3 are not evaluated on the same goal. Maybe the goal from Fig. 3 is particularly hard to reach? - Why no red curve in Fig. 3? (especially since HER does not get the best result on the Sliding task) - Appendix l. 456: "kicking" => sliding - It would have been interesting to see experiments combining reward shaping with HER where reward shaping is only used to "fine-tune" the reward near the goal state - Another potentially interesting experiment would have been to investigate if there could be additional benefits to also using "fake" goals that were not attained -- possibly sampled near an attained goal